# Universal scaling in real dimension

Giacomo Bighin[1], Tilman Enss [1] & Nicolò Defenu [2] ✉

The concept of universality has shaped our understanding of many-body physics, but is mostly limited to homogenous systems. Here, we present a study of universality on a non-homogeneous graph, the long-range diluted graph (LRDG). Its scaling theory is controlled by a single parameter, the spectral dimension $d_s$, which plays the role of the relevant parameter on complex geometries. The graph under consideration allows us to tune the value of the spectral dimension continuously also to noninteger values and to find the universal exponents as continuous functions of the dimension. By means of extensive numerical simulations, we probe the scaling exponents of a simple instance of $O(\mathcal{N})$ symmetric models on the LRDG showing quantitative agreement with the theoretical prediction of universal scaling in real dimensions.

Universality lies at the core of the modern theory of critical phenomena. With this concept one refers to the property of the scaling laws, observed in the vicinity of a thermal phase transition (PT) or of a quantum critical point (QCP), not to depend on the microscopic details of the model at hand[1,2]. In particular, in lattice systems with finite-range interactions the critical exponents are not affected by any modification in the interaction range or in the structure of the regular lattice.

One of the fundamental consequences of universality is the possibility of identifying continuous field theories, whose universal properties exactly reproduce experimental and numerical observations of actual lattice models for given values of the dimension $d$ and the symmetry properties of the order parameter[3–5].

The most celebrated example of universality in this context is represented by $O(\mathcal{N})$-symmetric field theories, which describe the critical properties of a wide range of physical transitions, such as finite temperature ferromagnetic, quantum anti-ferromagnetic, liquid-vapour, superfluid and superconducting transitions[2]. The microscopic action of $O(\mathcal{N})$ field theory reads

$$S[\boldsymbol{\varphi}] = \int d^d x \left\{ \frac{1}{2} \partial_\mu \boldsymbol{\varphi} \cdot \partial_\mu \boldsymbol{\varphi} + \frac{m^2}{2} |\boldsymbol{\varphi}|^2 + \frac{g}{24} |\boldsymbol{\varphi}|^4 \right\} \qquad (1)$$

where $\boldsymbol{\varphi}$ is an $\mathcal{N}$-component vector, $\mu$ runs over all the coordinates of the $d$-dimensional Euclidean space, while $m^2$ and $g$ are the couplings of theory, which may depend on the details of the microscopic model under study.

The model in Eq. (1) undergoes a spontaneous symmetry breaking (SSB) transition at a critical value $m^2 = m_c^2$. In the symmetry broken phase $m^2 < m_c^2$ a finite condensate or magnetization appears leading to a finite ground-state expectation value of the field operator $\boldsymbol{\varphi}$, i.e., $\langle \boldsymbol{\varphi} \rangle \neq 0$. As the square mass approaches the critical point $m^2 \to m_c^2$ the thermodynamic functions and observables display scaling behaviour, with scaling indices that are universal as they depend only on $d$ and $\mathcal{N}$ for any $g > 0$. Approaching criticality from a symmetric phase, the two-point correlation function often assumes the scaling form $\langle \varphi(x)\varphi(0) \rangle \approx \exp(-|x|/\xi)/|x|^{d-2}$, where the correlation length scales as $\xi \approx \lambda^{-\nu}$ and $\lambda$ is the control parameter that controls the proximity to perfect criticality $\lambda_c = 0$. The correlation length exponent $\nu$, as defined by the above equations, has been the subject of intense field theoretic studies, which led to its characterization on the entire real line $d \in \mathbb{R}$[6–8]. Most of the aforementioned field theoretic estimates accurately reproduce exact numerical simulations on regular lattices with integer dimensions[9,10]. In this perspective, it is fair to say that universality and critical scaling have been thoroughly scrutinized both via advanced theoretical tools and extensive numerical simulations, yielding a coherent quantitative description of universal properties on regular lattices.

On the other hand, complex systems, whose microscopic components occupy the sites of a non-homogeneous graph rather than a regular lattice, are also expected to display universality, but their scaling behaviour still presents several open questions. The importance of universal scaling on complex networks, and non-homogeneous structures in general, is becoming increasingly

[1]Institut für Theoretische Physik, Universität Heidelberg, 69120 Heidelberg, Germany. [2]Institut für Theoretische Physik, ETH Zürich, Wolfgang-Pauli-Str. 27, 8093 Zürich, Switzerland. ✉e-mail: ndefenu@phys.ethz.ch

relevant as atomic, molecular and optical (AMO) physics experiments push the control of Rydberg states to the single-atom level, allowing the construction of tuneable structures where non-homogeneity and strong correlations coexist[11]. At the same time, coherent Ising machines provide photonic realizations of large, programmable non-homogeneous networks operating near a PT[12].

The present work provides a relevant case study of a specific critical system whose universal behaviour is consistent with the following general conjectures:

S1. The universal exponents of a critical system on a disordered self-averaging graph in the absence of frustration depend only on the spectral dimension $d_s$ of the graph.

S2. The universal scaling as a function of $d_s$ coincides with that of a continuous field theory with $d = d_s$.

In statement S1, we designate graphs as self-averaging when their Laplacian spectrum remains independent of the specific disorder realization in the thermodynamic limit. This designation applies specifically to graphs with a well-defined spectral dimension, where the local and average definitions of the spectral dimension coincide, as elucidated in ref. 13. Furthermore, for these statements to be applicable, the microscopic model under investigation must possess an $O(\mathcal{N})$ field theory description in the low-energy (continuous) limit, in particular when its components are arranged on the sites of a regular lattice.

A more general version of statement S1 has already been discussed in the literature, since early studies indicated $d_s$ as the control parameter for universality on non-homogeneous structures[14,15]. However, general inhomogeneous structures may not have a well-defined global spectral dimension and thus may escape the universality scenario[13,16–19]. In this work, we construct a specific model, the long-range diluted graph (LRDG), whose spectral dimension is tuneable and well-defined in the thermodynamic limit. We then study an instance of universal scaling on the LRDG and discuss under which assumptions this is consistent with the conjectures S1 and S2.

Regarding statement S2, the correspondence between the universal properties in systems with equivalent spectral dimensions has been discussed in the case of long-range and glassy systems[20,21], but it has been found to be only approximate[22,23]. Our numerical study focuses on the whole range $d_s \in [2, 4]$ and supports the correspondence of the LRDG of spectral dimension $d_s$ with a continuous theory such as Eq. (1) with $d = d_s$. We argue that the LRDG may be a first example where this correspondence actually holds exactly.

Most numerical investigations of critical phenomena on complex networks focus on small-world networks, where critical fluctuations are Gaussian, and the critical indices correspond to the ones predicted by mean-field theory. For the study of critical behaviour within mean-field theory we refer the interested reader to Chap. 5 of ref. 24. In other words, the critical exponents of most statistical mechanics models on small-world networks correspond to the ones of lattice systems above the upper critical dimension and are not affected by network properties such as clustering coefficients, degree correlations and fractal dimension[25,26]. For a more extensive discussion on the Gaussian nature of critical correlations on small-world networks see Sec. IX of ref. 27.

However, correlated scaling behaviour is expected to occur on complex networks with gapless Laplacian spectrum. There, the density of states (DOS) $\mathcal{D}(\varepsilon)$ of the Laplacian spectrum displays power-law scaling at low energies ($\varepsilon \to 0$), leading to the definition of spectral dimension

$$\mathcal{D}(\varepsilon) \approx \varepsilon^{d_s/2-1}, \tag{2}$$

see also ref. 28 for a more rigorous definition in terms of the infrared singularities of a Gaussian model. An integer spectral dimension $d_s$ is

typical of regular lattices, where it coincides with the Euclidean and the fractal dimension $d_s = d = d_f$. The spectral dimension is commonly real on complex networks and its value controls the occurrence of SSB on these structures.

Indeed, for a discrete symmetry model $\mathcal{N} < 2$ on regular lattices, SSB occurs if and only if $d > 1$, while for continuous symmetries the corresponding condition is $d > 2$, as stated by the celebrated Mermin–Wagner theorem[29,30] and its inverse[31,32]. The extension of this result to non-homogeneous structures with $d_s > 1$ ($d_s > 2$) for discrete (continuous) symmetries has been described in refs. 14,15. The definition of spectral dimension itself displays some universal characteristics, such as the equivalence between Laplacian spectrum, vibrational spectrum and random walk definitions as well as the independence of the mass distribution in a Gaussian model[33]. Finally, the universal behaviour of $O(\mathcal{N})$ models on complex networks is determined solely by the spectral dimension in the $\mathcal{N} \to \infty$ limit and coincides with that of large-$\mathcal{N}$ Heisenberg ferromagnets[34,35].

All these results point to the spectral dimension as the natural control parameter for universality on complex networks, where it should play the same role as the Euclidean dimension on regular lattices. At present, however, there are no numerical simulations showing that the universal behaviour of critical theories on complex networks with given $d_s \in \mathbb{R}_+$ does indeed reproduce the scaling behaviour of continuous $O(\mathcal{N})$ field theories with $d = d_s$. This may be due to the lack of suitable examples of graphs with well-defined spectral dimension in the appropriate range $d_s \in [2, 4)$ where correlated scaling behaviour appears. Here, we intend to provide substantial evidence that the universal scaling of microscopic statistical mechanics models on non-homogeneous structures is indeed described by an appropriate quantum field theory (QFT) in real dimension. While we use the term QFT to refer to the model in Eq. (1), our studies always consider Euclidean spacetime and thus describe only classical PTs[2].

## Results

The microscopic model of our choice is a self-avoiding random walk (SARW) that jumps between the vertices along the edges of a 2D long-range dilute graph (LRDG). We generalize the SARW definition of ref. 36 to a simple graph. An $N$-step self-avoiding walk on a graph $G$ is a set of $N + 1$ points $R_0 = 0, R_1, R_2, ..., R_N$ in $G$ with $|R_{i+1} - R_i|_G = 1$ and $R_i \neq R_j$ for all $i \neq j$, where $|\cdots|$ is the chemical distance defined on the graph $G$. A probability measure is defined on the set of all $N$-step self-avoiding walks by assigning an equal probability to each such walk. According to the previous definition the SARW is not a Markovian process and is also not stricto sensu a stochastic process[36], at variance with the case of the true self-avoiding walk[37].

The SARW is one of the simplest systems in statistical physics that exhibits correlated critical behaviour. We define $R_N$ as a measure of the extension of the SARW—such as the end-to-end distance or the gyration radius—after having performed $N$ steps. Then, in the $N \to \infty$ limit one has

$$\langle R_N^2 \rangle \approx A N^{2\nu} \tag{3}$$

where $A$ is a non-universal constant and $\nu$ is a critical exponent, the celebrated Flory exponent. Moreover, in the same limit one can also define the critical exponent $\gamma$ via

$$p_N \approx e^{-\mu N} N^{\gamma-1} \tag{4}$$

where $p_N$ is the survival probability for an $N$-step SARW and $\mu$ is the non-universal connective constant of the lattice. Remarkably, such a simple model is also able to reproduce very realistically various aspects of polymers in solution[38,39].

We use the SARW as a prototypical example of universal behaviour because it has three characteristics that are highly valuable for our studies:

(i) It allows for efficient and reliable numerical simulations, which have led to highly accurate estimates of the correlation length exponent $\nu = 0.587597(7)$ on regular lattices in $d = 3$[36,40,41].

(ii) Its free energy can be exactly related to that of $O(\mathcal{N})$ field theories in the $\mathcal{N} \to 0$ limit[42–44], yielding a direct continuum counterpart for its universality class[45].

(iii) Scaling arguments can be used to derive the celebrated Flory estimate[46,47]

$$\nu = \begin{cases} 3/(d+2) & \text{if } d < 4 \\ 1/2 & \text{if } d \geq 4 \end{cases} \qquad (5)$$

for the correlation length exponent. These scaling arguments do not only reproduce the mean-field result in $d \geq 4$, but also give the exact result $\nu = 3/4$ in $d = 2$[48]. As a consequence, Eq. (5) gives a very reliable estimate for the universal exponent $\nu$ in the whole dimensional range $d \in [4, 2]$ with only a 2.1% deviation from the numerical value in $d = 3$[45].

The critical exponent $\nu$ describes the diffusion properties of the SARW. For $d > 4$ the model is purely diffusive as for the conventional (Markovian) random walk[49], while in the range $2 < d \leq 4$ the system behaves super-diffusively. Finally, in $d = 1$ it can be exactly proven that the scaling is purely ballistic[50]. In the following, we will show that the LRDG continuously interpolates between the diffusive behaviour in large dimensions and the purely ballistic scaling in $d = 1$.

On the other hand, the choice of the LRDG builds on our previous studies on the long-range random ring (LR³)[51], where it was shown that the spectral dimension of a 1D lattice can be tuned in the range $d_s \in [1, +\infty)$ by including additional links distributed according to a power-law distribution as a function of the link length $r$, i.e., $P = r^{-\rho}$. With respect to the LR³, its 2D generalisation, i.e., the LRDG, has two main advantages:

(i) The disorder contribution to the spectral dimension $d_s$ is irrelevant, as we will argue below, suppressing the interplay between disorder and critical correlations and thus simplifying the comparison with the homogeneous model.

(ii) In the $\sigma \to \infty$ limit, the long-range disorder contribution vanishes and the LRDG reproduces a regular 2D lattice, where the SARW critical exponents are known exactly, providing a straightforward benchmark for our investigations.

According to our statement S1, the universal scaling of a microscopic system whose components occupy the edges of the LRDG coincides with the continuous theory in Eq. (1) at $d = d_s$. Thus, for the SARW on the LRDG we expect the correlation length exponent $\nu(d_s)$ to coincide with that of the $O(\mathcal{N})$ model in the $\mathcal{N} \to 0$ limit, which is close to the Flory estimate Eq. (5). The existence of the critical point both for the LRDG with dimension $d_s$ and the continuous model with $d = d_s$ is granted by the generalization of the Mermin–Wagner theorem and its inverse to graphs[13–15,52]. While the numerical study in Refs. 53,54 is consistent with the mathematical theorems, a numerical proof that the universality on graphs of dimension $d_s$ really corresponds to that of the continuous theory in $d = d_s$ is still missing.

Let us now explain in detail how the LRDG is constructed: one starts by considering an $L \times L$ Euclidean 2D lattice, and builds a graph in which each vertex corresponds to a point in the lattice, and each edge connects first neighbours. We impose periodic boundary conditions at the borders. Subsequently, we consider all possible edges connecting non-first-neighbours and add them to the graph with probability $P = r^{-\rho}$ for Euclidean distance $r$ between the two sites (see Fig. 1). For convenience we rewrite our decay exponent as

$\rho = 2 + \sigma$ and from now on we consider the case $\sigma > 0$. Note that the in the limiting case $\sigma = -2$ we obtain the fully connected—i.e., complete—graph, whereas for $\sigma \to \infty$ we recover the underlying Euclidean lattice with no additional edges. Several analytic expressions for the graph properties of LRDG and its generalizations are known[55].

One may expect that the equivalent real dimension $d_s$ of the LRDG is related to $\sigma$ via the relation

$$d_s = \begin{cases} \frac{(2-\eta)d_{\text{latt}}}{\sigma} & \text{if } \sigma < 2 - \eta, \\ d_{\text{latt}} & \text{if } \sigma \geq 2 - \eta, \end{cases} \qquad (6)$$

where $d_{\text{latt}}$ is the (Euclidean) dimension of the underlying lattice, i.e., $d_{\text{latt}} = 2$ in our case. The quantity $\eta$ encodes the contribution arising due to disorder correlations. In fact, averaging the adjacency matrix of the LRDG graph over the probability distribution $P$ yields a conventional fully connected long-range system[11,56]. This procedure is referred to as annealed disorder average (ADA) and leads to the spectral dimension $d_s = 2d/\sigma$ for $\sigma < 2$, i.e., $\eta = 0$ as expected. On the other hand, when the disorder is properly accounted for and the average is taken directly on the spectrum, i.e., a quenched disorder average (QDA), one finds for the LR³ model $\eta \neq 0$[51]. However, previous studies of the LRDG seemed to be consistent with $\eta = 0$ also for QDA[53].

The experienced reader may notice the similarity between our Eq. (6) and the effective dimension relation discussed in refs. 20–22,57 for both fully connected and disordered systems. However, the effective dimension relation involves a model-dependent anomalous dimension and has been shown to be only approximate[22,23]. In the following, we will argue that the scaling is our model is governed by Eq. (6) rather than the traditional effective dimension for clean, fully connected long-range systems.

In particular, given the universal properties of the spectral dimension[33], we do not expect the form of the underlying lattice to affect the result in Eq. (6) as much as it should not affect the universal scaling of any statistical mechanics model. Nevertheless, a modification of our construction based on a triangular or hexagonal lattice may lead to a significant change in the subleading finite-size scaling behaviour, thus improving the performance of the numerical algorithm.

Once the graph is constructed, one has to work out a way of generating SARWs on top of it. Whereas exact enumeration approaches are possible[58], critical exponents and related quantities are better investigated by means of stochastic Monte Carlo (MC) approaches. Recently, high-precision MC tools for SARWs have been developed, essentially sophisticated refinements of the pivot algorithm[41,59], allowing the determination of the Flory exponent to 6 significant digits, one of the most precise quantities ever measured in statistical mechanics[41]. Remarkably, this class of algorithms disproves Sokal's conjecture[39] that no effectively independent SARW can be generated from an existing one faster than $O(N)$ for walks of length $N$.

Unfortunately, these improved methods make use of the underlying symmetries of the Euclidean lattice and cannot be extended to the LRDG. Therefore, for the present investigation, we use an extension of the simpler 'slithering-tortoise' algorithm[60], a dynamic local Monte Carlo algorithm based on the simple idea of constructing an ergodic process that moves through the space of all SARW configurations on a given lattice using two simple moves that can be easily adapted to our case. The moves consist simply of fixing one end of the SARW at one vertex in the graph, and start by considering a 0-length walk. The '+' update will then extend the SARW by one step, selecting a candidate with probability $P(+)$, while the '-' update will reduce the length of the SARW by one step with probability $P(-)$. In case an update leads to a self-intersection or tries to shorten a 0-length walk, the move is rejected—a null transition—and the old configuration is counted again. The following choices for the acceptance probability of each

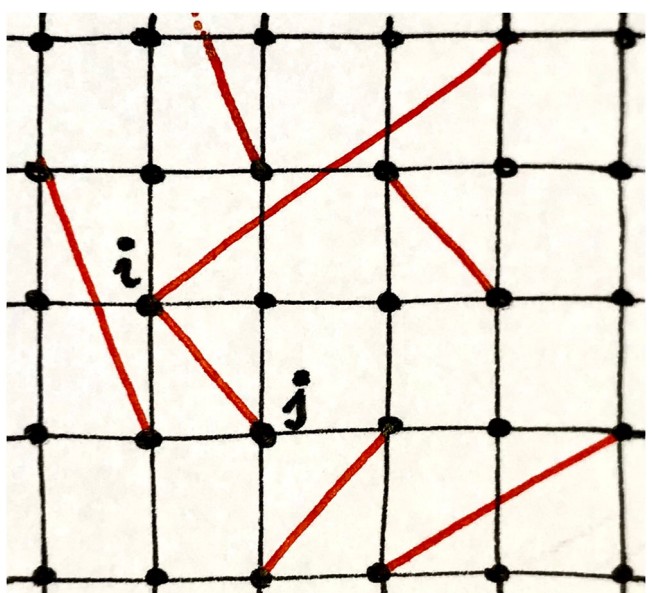

**Fig. 1 | Pictorial representation of the long-range diluted graph (LRDG).** The complex network has a $d = 2$-dimensional square lattice as its backbone, with additional long-range links (marked in red) that are switched on with probability $p_{ij} = r_{ij}^{-d-\sigma}$, which depends as a power law on the distance $r_{ij}$ between sites $i$ and $j$. The local connectivity $k$ (in the figure $k = 6$ at site $i$ and $k = 5$ at site $j$) follows a broad distribution.

update satisfy the detailed balance condition,

$$P(+) = \frac{q\beta}{1+q\beta}, \quad P(-) = \frac{1}{1+q\beta}, \tag{7}$$

where $q$ is the number of neighbours in the graph *before* proposing the update in $P(+)$, and after (potentially) accepting the update in $P(-)$. Furthermore, the parameter $\beta$ in the transition probabilities plays the role of a pseudo-chemical potential that regulates the average length of the walk; it can be easily demonstrated[60] that a walk $\omega$ will be sampled by this update scheme with probability

$$\pi(\omega) \propto \beta^{|\omega|} \chi_{\text{SARW}}(\omega) \tag{8}$$

where $|\omega|$ is the length of the walk, and $\chi_{\text{SAW}}(\omega)$ equals 1 for a SARW and 0 for a self-intersecting walk. In practice, $\beta$ can be tuned empirically until the average length of the walks reaches the desired value. It is immediately apparent that the algorithm is ergodic for SARWs while also generating samples that are largely correlated with an autocorrelation time $\tau \sim \langle N \rangle^2$.

In our simulations, we considered Euclidean lattices up to $L \times L = 256 \times 256$ as the backbone for the construction of an LRDG graph, with $\sigma$ in the range $\sigma \in [1, 2]$. Our MC simulations consist of 128 averages over different realisations of the LRDG with the same effective dimension, while also averaging over 128 different walks on the same graph, also with different randomly chosen starting sites. After an initial thermalisation phase of $5 \times 10^6$ MC steps, we perform $20 \times 10^6$ MC steps, sampling the end-to-end radius—using the underlying Euclidean metric—and the survival probability as a function of the SARW length $N$.

First, we determine the spectral dimension $d_s$ of each different realisation of the LRDG by computing the low-energy spectrum of the graph Laplacian[61] associated with each graph, and then derive the spectral dimension via a procedure detailed in Supplementary Note I. A similar analysis has been performed for the 1D case in ref. 51 comparing it against other approaches; we verified that the Laplacian

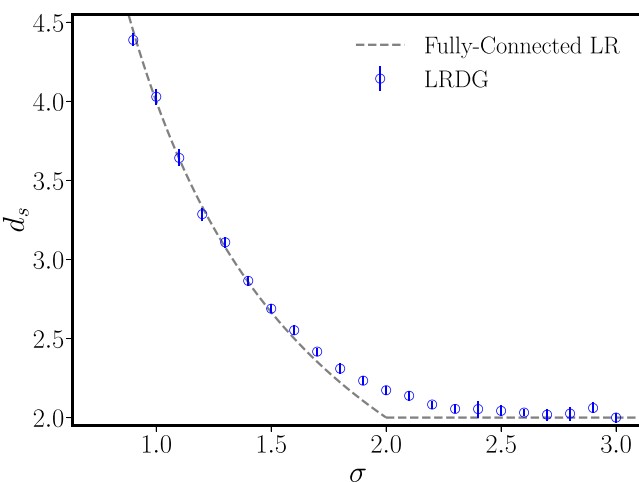

**Fig. 2 | Spectral dimension $d_s$ of the LRDG graph.** The value of $d_s$, obtained from the scaling of the Laplacian spectrum (blue circles), is compared with the analytical result obtained for fully connected long-range interactions (grey dashed line). The deviation around $\sigma = 2$ is probably a consequence of logarithmic corrections, see the discussion in the main text. The symbol contains vertical lines representing uncertainty. Each round symbol is obtained by averaging over the values of $d_s$ obtained by fitting the first 24 low-lying eigenvalues of the LRDG Laplacian spectrum, the uncertainty is obtained as the standard deviation, see the Supplementary Note I for more information.

spectrum method provides the best estimates also in the LRDG case. The outcome of the numerical analysis presented in the Supplementary Note I is reported in Fig. 2, where the spectral dimension of the LRDG model is compared with the equivalent result for fully connected long-range interactions reported in Eq. (6).

The numerical estimates of $d_s$ obtained for the LRDG (blue circles in Fig. 2) do not display any substantial shift away from the ADA prediction and are overall consistent with a correction $\eta = 0$ for $\sigma < 1.5$. At larger values of $\sigma$ the deviation from the ADA estimate $d_s = 2d/\sigma$ grows and reaches a maximum at $\sigma \approx 2$. From this, one would at first glance conclude that $\eta \neq 0$ also for the LRDG, as found already for the LR³ model[51]. However, while the LR³ produced a correction $\eta > 0$, the deviation observed in Fig. 2 would yield a correction $\eta < 0$. Thus, the deviation observed for $1.5 \lesssim \sigma \lesssim 2.5$ in Fig. 2 is more likely the result of a systematic error produced by logarithmic corrections, which emerge near the marginal value $\sigma = 2$. Similar corrections appear in most long-range models[56] and have proved difficult to capture[21,22] due to the large sizes needed in the simulations[62]. Further evidence of the finite-size origin of the correction $\eta < 0$ in $d = 2$ is found in Fig. 2 in the Supplementary Information, where the numerical estimate of $d_s$ is reported as a function of the system size. The two panels on the left of this figure clearly show the emergence of nonlinearities in the finite-size scaling (not included in our fitting function), which increase the numerical estimates of $d_s$ at $\sigma \gtrsim 0$. Previous numerical studies on a related model, which is expected to be in the same universality class of the LRDG based on the arguments of ref. 33,63, were also consistent with $\eta = 0$[53].

Based on the previous arguments and in agreement with ref. 53 we assume in our subsequent analysis that the spectral dimension of the LRDG follows Eq. (6) with $\eta = 0$. Since our analysis will mostly focus in the neighbourhood of $d_s = 3$, which falls below $\sigma = 1.5$, the quantitative analysis of the possible correction $\eta \lesssim 0$ will be left to future work.

In order to validate the universality of scaling phenomena on the LRDG graph, we computed the correlation length exponent of the SARW on the graph. The results of this analysis are reported in Fig. 3 and compared with the theoretical expectation obtained by the Flory theory, replacing the integer dimension $d$ with the spectral dimension $d_s$ in Eq. (5). The MC data fall neatly on the theoretical curve in the

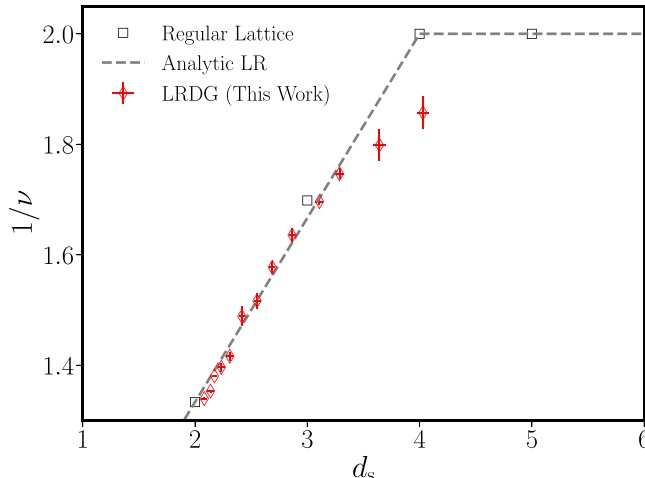

**Fig. 3 | Inverse correlation length exponent $1/\nu$ as a function of the spectral dimension $d_s$.** The value of $1/\nu$, obtained from Monte Carlo simulations (red diamonds), is compared with analytical and numerical results in integer dimensions (black squares) and with the Flory theory prediction Eq. (5) with $d = d_s$ (grey dashed line). Uncertainties on the regular lattice are not shown as the values in $d = 2, 4$ are exact and the one in $d = 3$ has a precision better than 0.01%. The symbol contains vertical and horizontal lines representing the uncertainty. Each diamond symbol is obtained by fitting the gyration ratio of the SARW for the three largest sizes of the LRDG and taking the average. The uncertainty is obtained as the largest deviation between the aforementioned values, see the Supplementary Note II for more information. The MC uncertainty of the square boxes is smaller than the size of the symbol.

whole range $d_s \in [2, 3.5]$ providing a very strong indication of universality in the LRDG.

In the region $d_s \approx 4$ the agreement is poorer, probably due to the appearance of finite-size logarithmic corrections close to the mean-field threshold[64]. Indeed, numerical studies at and above the upper critical dimension ($d_{uc} = 4$ in our case) are plagued by logarithmic and finite-size corrections, which need to be explicitly accounted for in order to reproduce the expected universal predictions[65–67]. The impact of these logarithmic corrections on our numerical analysis is evident in the study of the curve collapse that we use to determine the universal scaling region, see the Methods section. For a modern discussion of the impact of logarithmic correction close to the upper critical dimension and the difficulties they produce in matching theory and numerical simulations, see ref. 68.

Building upon the earlier discussion, we establish the first conjecture underlying our interpretation of the findings presented in Fig. 3:

C1. We propose that the deviation observed between the numerical data for the LRDG (represented by red diamonds in Fig. 3) and the analytical curve in Eq. (5) at high values of $d = d_s$ stems from finite-size corrections inherent in the numerical simulation. This disparity is expected to diminish as the system approaches the thermodynamic limit.

The subsequent discussion is based on this assumption.

The attentive reader may be surprised by the good agreement obtained between our numerical estimates and the theoretical curve at $d_s \approx 2$, in light of the (previously discussed) systematic corrections affecting $d_s$ in this range, see Fig. 2. However, the justification is rather straightforward. Since the critical exponents on the LRDG depend only on the spectral dimension, it is reasonable to expect that the numerical estimates of the critical exponents on the finite graph probe the finite-N corrected spectral dimension, rather than the expected thermodynamic limit result. Thus, when

reporting the numerical estimate of the correlation length exponent as a function of the numerical $d_s$, we observe the cancellation of two different systematic corrections.

To aid the ensuing discussion, we introduce an additional assumption concerning the scaling of finite-size corrections:

C2. We posit that the compensation between finite-size corrections in the spectral dimension $d_s$ of the LRDG and in the critical index $\nu$, as manifested in the lower $d_s$ range of Fig. 3, will persist consistently at large $N$.

This anticipation aligns with the natural expectation, as outlined in C1, that the spectral dimension $d_s$ will decrease with increasing $N$. At the same time, consistent with the overarching framework, the critical index $\nu$ is expected to increase. This additional conjecture C2 complements our analytical framework.

The evolution of the correlation length exponent as a function of $d_s$ together with assumptions C1 and C2 demonstrates that the universality on the LRDG network depends only on the low-energy spectrum, since it is in agreement with the Flory prediction, which is derived solely from low-energy scaling theory. Interestingly, this correspondence also extends to the case of self-avoiding Lévy flights (SALF) in 1D, where the length of the jump is distributed according to a power-law distribution, whose critical exponents have been shown to be consistent with the Flory estimate in ref. 69.

## Discussion

We have presented a numerical study of self-avoiding random walks (SARW) on a simple graph with tuneable spectral dimension. The graph structure is depicted in Fig. 1, which we refer to as long-range diluted graph (LRDG). As a function of the decay exponent $\sigma$ of the power-law probability distribution the graph realizes all possible spectral dimensions $d_s \in [2 + \infty)$, similarly to the case of fully connected long-range interacting systems[11,56]. However, the LRDG is merely a simple graph, i.e. a graph whose edges all have the same value. Therefore, it provides a proper representation of a nearest neighbour model on a complex topology[51].

In this respect, the model has already been used to study several properties of complex systems and spin glasses as a function of a continuous dimension[20,70–74]. Here, we employ the LRDG to investigate universal scaling on complex topologies. However, in contrast to previous studies on spin glasses, our model has no frustration and is perfectly self-averaging, as we have already shown in ref. 51 for the 1D version.

The self-averaging nature of the LRDG is crucial to ensure proper universality. Indeed, it is known that pathological graphs exist where the local expectation of the spectral dimension does not coincide with that observed on average[75], leading to universality violations and in particular to the absence of SSB in $d_s > 2$[13,16–19].

Previous numerical studies of non-frustrated structures have focused on exactly solvable models[33,35,76] or on the existence of SSB[53,54]. Here, however, we report for the first time the complete curve of a critical index in the entire spectral dimension range $d_s \in [2, 4]$, where the universality is expected to be correlated, i.e. not mean-field. The agreement between the numerical findings (red diamonds in Fig. 3) and the theoretical prediction in Eq. (5), which has been derived by simple scaling arguments without any information about the structure of the LRDG, is a strong indication of universality on the LRDG graph. Moreover, the theoretical prediction in Eq. (5) has already been shown to reproduce the critical scaling of SALF[69], whose scaling theory is similar to that of our model.

While Fig. 3 strongly suggests the universality of SARW on the LRDG graph, it is essential to acknowledge that this conclusion relies on the two key assumptions, C1 and C2. These assumptions effectively

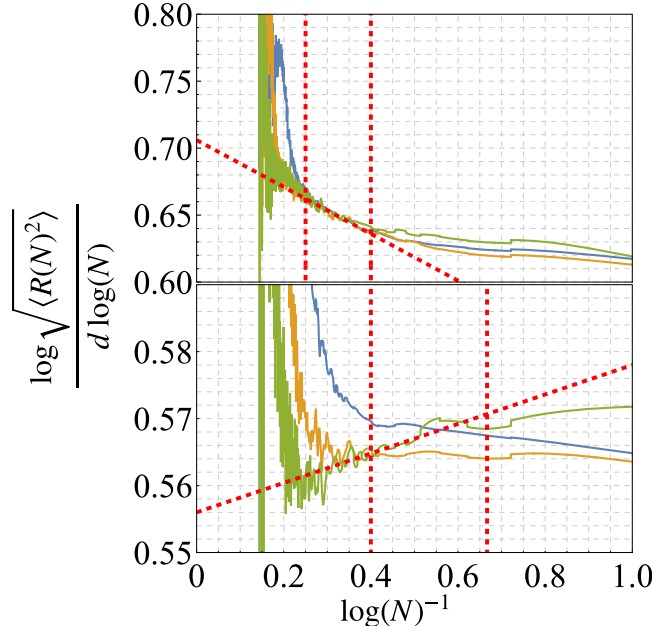

**Fig. 4 | Logarithmic derivative of the average spatial extent of the random walk.** The plot displays the logarithmic derivative of the average spatial extent of the random walk as a function of the inverse logarithm of the number of steps $N$. Data shown for $\sigma = 1.8$ and $\sigma = 1.1$, corresponding to $\rho = 3.8$ (top) and $\rho = 3.1$ (bottom), for different lattice sizes: $L \times L = 64 \times 64$ (blue), $128 \times 128$ (yellow), $256 \times 256$ (green). The collapse region is well established for larger values of $\rho$ and corresponds to a regime where scaling corrections and finite sizes effects are negligible, so that one can define a `confidence region', as bracketed by the vertical red dashed lines, and extrapolate from there the critical exponent.

confine any disparities between the theoretical framework and numerical observations to finite-size corrections. Additionally, a rigorous confirmation of statements S1 and S2 necessitates the consideration of further assumptions:

C3. Any critical system whose low-energy description adheres to an $O(\mathcal{N})$ field theory obeys universality on the LRDG graph, where its critical exponents exclusively depend on $d_s$.

C4. Our findings can be extended to any graph possessing a self-averaging spectrum.

C5. The universality class of fully connected long-range models studied above differs from that of diluted models, such as the LRDG.

Consequently, a comprehensive validation of the conceptual framework outlined in the introduction and encapsulated in statements S1 and S2 necessitates the justification of all the hypotheses C1–5.

In order to verify hypotheses C1 and C2 a more extensive numerical campaign ought to be performed as well as an in-depth theoretical derivation of proper fitting functions for the SARW model in the non-homogeneous setting. Further on, the study of different $O(\mathcal{N})$ universality classes such as the Ising ($\mathcal{N} = 1$) or XY ($\mathcal{N} = 2$) is envisaged. In those cases no accurate expressions such as Eq. (5) for the critical indices is available. Thus, a consistent comparison between the universal properties of the microscopic model, obtained via numerical simulations, will also require the determination of the universal scaling for the corresponding continuous field theory, see Eq. (1). Several theoretical tools are available for this purpose including conformal bootstrap[8], functional RG[77] and perturbative expansions[10].

With the progress in the manipulation of Rydberg atoms confined into optical tweezers, it will be possible to realize quantum simulations of complex systems on non-homogeneous topologies[78,79], similarly to the ones performed in fully connected models[80,81]. This will certainly boost the importance of the study of universality on non-homogeneous structures.

As a future perspective, we mention the possibility of studying other properties of the SARW on the LRDG. These studies will mainly focus on other (possibly) universal quantities such as the critical exponent $\gamma$ or the value of the critical free energy, which also show some form of universality[82]. The effect of non-frustrated disorder on these quantities will be crucial to fully understand universality on non-homogeneous graphs.

Our studies pave the way for the use of diluted models to perform numerical simulations of universal physics on fully connected long-range systems. Indeed, they both share the same scaling properties and, according to our studies, they are likely to exhibit the same universal phenomena. Since diluted models have a reduced degree of connectivity and, even, sparse coordination matrices[51,53], they can be used to perform large-scale quantum simulations of strongly interacting models with non-analytic spectral properties with a fraction of the effort.

## Methods

The present method section summarizes how we obtained the estimate for $\nu$ from the numerical simulations, while more details about the determination of $\nu$, as well as the procedure for the determination of $d_s$, are reported in the Supplementary Information. After having performed an MC simulation and obtained an estimate for the end-to-end distance $\langle R^2(N) \rangle$, the main difficulty encountered by the numerical analysis derives from the presence of systematic corrections to the $\langle R^2(N) \rangle$ curve both at small and large values of $N$. The small-$N$ deviations naturally occur due to subleading corrections $O(N^\omega)$ with $\omega < 2\nu$, which modify the leading-order scaling reported in Eq. (3). At large values of $N$, on the other hand, the universal scaling is disrupted by the finite-size of the LRDG graph, which affects long walks that can wind around the boundary of the system.

In order to extract the correlation length exponent from the scaling of $\langle R^2(N) \rangle$, one needs to identify a suitable region of $N$ where one can reasonably apply the relation in Eq. (3). This universal region can be identified by observing the collapse of different curves $\langle R^2(N) \rangle$ for different system sizes $L$. This analysis is reported in Fig. 4, for both $\sigma = 1.8$ and $\sigma = 1.1$. In the upper panel the collapse in rather evident in a wider region of N, while in the lower panel the collapse region is harder to determine. The universal window is visually identified as the region where the three curves overlap and exhibit a linear behavior. For all $\sigma \gtrsim 1.5$ the universal window roughly corresponds to the region where the disagreement between the three curves remains below 5%. The same analysis has been repeated for all values of $\sigma$ resulting in very extended and well-defined fitting regions for all $\sigma \gtrsim 1.5$, similarly to what is reported in the upper panel in Fig. 4. For $1.5 \gtrsim \sigma \gtrsim 1.1$ the collapse region is less extended but still pronounced enough to obtain reliable $\nu$ estimates, at least up to $d_s \simeq 3.5$, see the lower panel in Fig. 4. Finally, in the region $\sigma \lesssim 1$ (corresponding to $d_s \gtrsim 4$) it is not possible to identify a clear universal region and, accordingly, the $\nu$ estimates become less precise, see the Supplementary Note II. As a consequence, the agreement between our data and the theoretical line is poorer in this region, see Fig. 3.

Once the universal regime has been determined following the aforementioned procedure, the computation of the Flory critical exponent $\nu$ is rather straightforward. The numerical data in the universal window are interpolated via a linear function of the inverse walk length and then extrapolated to $N \to \infty$, for more details see the Supplementary Note II.

## Data availability

The data displayed in the figures are available on figshare at the https://doi.org/10.6084/m9.figshare.24898476. All the raw data that support the findings of this study are available from the corresponding author upon request.

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

## Acknowledgements

We gladly acknowledge Giacomo Gori for his help in determining the spectral dimension from the numerical data. N.D. acknowledges stimulating discussions with Mehran Kardar. This work is supported by the Deutsche Forschungsgemeinschaft (DFG, German Research Foundation) under Germany's Excellence Strategy EXC2181/1-390900948 (the Heidelberg STRUCTURES Excellence Cluster) and by the Swiss National Science Foundation (SNSF) under project funding ID: 200021 207537. The authors acknowledge support from the state of Baden-Württemberg through bwHPC. This research was supported in part by grant NSF PHY-1748958 to the Kavli Institute for Theoretical Physics (KITP).

## Author contributions

G.B. wrote the numerical simulation Monte Carlo code and ran the numerical simulations. T.E. provided theoretical as well as methodological insights. N.D. provided methodological insights, organized the research questions and supervised the study. All authors substantially contributed to the manuscript, to the analysis and to the interpretation of the results.

## Competing interests

The authors declare no competing interests.
