## [Peer Review File · Nature Communications]

REVIEWER COMMENTS

Reviewer #1 (Remarks to the Author):

This is a report on the manuscript "Universal scaling in fractional dimension" that has been submitted for publication to Nature Communications.

The authors perform a numerical study of the critical behaviour of a statistic model, the Self-avoiding Random Walk (SARW), when defined on instances of the 2D Long Range Diluted Graph (LRDG), a probabilistic ensemble of networks of which one can tune the spectral dimension d_s as a function of a continuous parameter ρ . For various values of ρ , the authors estimate both the spectral dimension associated to the average spectrum of such graphs, and the correlation length critical exponent ν of the SARW. The resulting scatter relation $\nu(d_s)$ is therefore compared with the conjectured Flory theoretical relation $\nu(d)$, and with the known value for $\nu(d=3)$ (estimated by means of numerical simulations in the cubic lattice). The authors conclude (if I have correctly understood) that, for this model (the SARW), and for this graph family (the LRDG), the critical behaviour is solely governed by the spectral dimension d_s in the sense that, to some extent, it is consistent with that of the Euclidean lattice in d_s dimensions when d_s is integer and, in general, with the Flory relations, equation (5) with $d=d_s$. In matter of critical behaviour, d_s would then play the role of the (broadly understood) spatial dimension d .

In my opinion (and considering that I am no longer an updated expert in critical phenomena on complex networks), the results presented in the manuscript are original and relevant as they represent a further case of study in the analysis of critical phenomena in complex networks. The methods are unambiguously explained (yet, see below). The article works as well as an exhaustive bibliographic review.

However, I would ask the authors to elucidate some fundamental (and, in my opinion, unavoidable) questions that I pose below. Afterwards, I will suggest some minor, formal changes, in the hope that they could help improving the article's usefulness.

*

1) One of my main doubts regards the discussion on the spectral dimension in equation 6 and its equivalence with the "equivalent fractional dimension". On the one hand, the spectral dimension of the LRDG may have the form in equation 6, where η is something characteristic of the graph. In this sense η is, of course, independent on the eventual statistical model defined on the graph.

On the other hand, at least according to references 70,69,59,[Baños et al, Phys. Rev. B 86, 134416 (2012)], the equivalent fractional dimension, say D , corresponding to A GIVEN statistical model, has been conjectured to assume the approximated form as in equation 6 (with $d_s \rightarrow D$) with a model-dependent η , which is actually conjectured to be the anomalous scaling exponent η of the short-range equivalent model in a D -dimensional regular lattice (for ρ close enough to ρ_D such that $D(\rho_D)=D$).

Thus, whatever the spectral dimension d_s of an ensemble of networks is, the critical behaviour of a model defined on instances of such networks has been conjectured to coincide with that of the same ("short-range") model in D spatial dimensions, where D is related to ρ through a self-consistent relation for D (equation 6 with $d_s \leftrightarrow D$ and $\eta \leftrightarrow \eta_{sr}(D)$) which is actually model-dependent (since $\eta_{sr}(D)$ is a different function for different models). This is the sense of the $\eta=3-\rho$ argument below equation (3) of ref. 69; of the $\psi=\rho-d$ argument in equation (8) of ref. 46; of the free energy scaling arguments in reference [Baños et al, Phys. Rev. B 86, 134416 (2012)].

If I have well understood, this is contradictory with the thesis of this manuscript, claiming, instead, that

$$D = d_s$$

for all models. In this interpretation, $D=d_s$ holds always, and η in equation 6 is to be understood as a model-independent deformation of the spectral dimension, induced by the quenched disorder, with respect to that in the Annealed Disordered Average (ADA) that would lead to $\eta=0$.

(By the way, if this is true, I believe that, for the "mode" Nature Communications reader, it should be needed to explain what it is precisely meant by "averaging" the instances of a graph.)

In other words: there is known evidence of the fact that D is different from $2d/(\rho-d)$ in dilute long-range graphs with and without disorder, specially when $2d/(\rho-d)$ is close to d . But, while in references 69, 46 this difference is attributed to the fact that $D = d(2-\eta_{sr}(D))/(\rho-d)$, in the present reference it is claimed that $D=d_s$, and that $d_s = d(2-\eta)/(\rho-d)$. Both cases lead to the same

equation for D as a function of ρ . The main difference is that, in the former interpretation, η_{sr} is model-dependent, while in the present one it depends on the graph model only, and it is independent on the statistical model.

Is my summary correct?

Only in this is correct I understand: the discussion in the paragraph that hosts eq. (6); the fact that η is (solely) determined in the analysis of FIG. 2 by the graph properties; the main message of the article (claiming, if I have well understood, that d_s determines the critical behaviour of an equivalent "short-range" model at $D=d_s$ spatial dimensions for all models).

Assuming that I have understood correctly, I would suggest these two different interpretations to be stressed and discussed more clearly, and the facts in favour and against both hypotheses exposed. If I have not understood correctly, I would suggest to make the exposition more clear at this regard.

2) The authors claim that the spectral dimension d_s of the 2D LRDG is as in eq. (6) with $\eta=0$, despite the numerical evidence against this hypothesis for σ close enough to 2. The authors attribute the numerical difference with respect to $\eta=0$ to a finite- N effect. They argue, first, that an $\eta < 0$ would otherwise imply that the anomalous dimension η were < 0 for the (any?) model defined on this graph (if it is meant "any", why then do the authors refer to the Ising model in particular?), hence assuming that η is the anomalous dimension of the statistical model defined on the graph, which seems contradictory with the articles' thesis I summarised in point 1): i.e. η being model independent.

I would say, in any case, that the connection between (A) η in equation (6) (which seems referred to the spectral dimension of a (model-free) ensemble of networks) and (B) the critical exponent of a specific statistical model defined on these networks, needs to be explicitly stated and carefully explained, specially for non-specialist readers.

3) The authors claim, furthermore, that $\eta=0$ in eq. 6 is consistent with reference 46. However, the graph ensemble analysed in reference 46 is different from the one at hand since, in the former, the number of links is set to be constant and independent on ρ . This should be mentioned in the second paragraphs of sec. III and in the paragraph below eq. (6), or reference 46 removed in this discussion.

As far as I can see, at the end of the day there is no evidence of the fact that the d_s of the 2D LRDG is as in eq. (6) with $\eta=0$. I would suggest to consider not to conjecture much on this point since, in any case, the subsequent comparison with ν_{3d} is for a value of ρ for which $\{\eta=0 \text{ or } \eta \neq 0\}$ does not make any significant difference.

As a final comment, the estimation of d_s in reference 46 may be subject to significant biases, since the procedure used to estimate d_s in ref. 46 is not systematic, in the sense of not being computed through an unambiguously defined automatic procedure (see my above question in point (7)).

4) As the authors perfectly know, previous works have analysed the critical behaviour (value of critical exponents) of statistical models on a lattice of Euclidean dimension D , with that of the same models on both diluted and fully connected "long-range" networks (i.e. on diluted stochastic ensembles of networks with power-law decaying bond probability, and on fully connected networks with power-law decaying coupling weights).

The goal of these publications (see a review in ref. 70) is to see whether there is a value of ρ (equivalently, σ) for which all the independent critical exponents of the model on the graph are consistent to those of the short-range versions of the model in D spatial (Euclidean) dimensions. For different models, with or without disorder, the long \leftrightarrow short range analogy has been observed to hold for some values of ρ only.

In other terms: the universal relation [regular short range lattice on D dimensions] \leftrightarrow [long-range graph with power ρ , i.e., WITH $d_s=d_s(\rho)$] has been observed to be but an approximation in previous studies. This would suggest that, in general THERE IS NO VALUE OF d_s such that [short range in D] \leftrightarrow [long-range in $d_s=D$]. To what extent these results contradict the claim for universality of the present article?

I am probably missing this points since I am no longer updated on the topic. I would suggest nevertheless that the authors explain better the picture so that it is fully useful to non-expert readers.

5a) Perhaps my main doubt regards the results in FIG. 3.

First, the authors find agreement between the Flory $\nu_F(ds)$ and the numerical $\nu_{num}(ds)$ precisely in the region $ds \sim 2$ in which the numerical estimation of the spectral dimension ds is believed to suffer from finite-N effects. If the estimation for $ds_{num}(\rho)$ in FIG. 2 is unreliable for $\rho \sim 4$, how can the estimation for $\nu_{num}(\rho)$ vs $ds_{num}(\rho)$ be significant precisely for $\rho \sim 4$?

In other words: the red diamonds in Fig. 3 are, if I have well understood, a scatter plot with

$1/\nu_{num}(\rho)$ vs $ds_{num}(\rho)$

Now, what would happen if the authors plotted

$1/\nu_{num}(\rho)$ vs $ds(\rho, \eta=0)$

where $ds(\rho, \eta=0)$ is as in equation 6, with $\eta=0$? If the hypothetical scatter points in the last equation deviated significantly from the Flory estimation for $ds > 2$, wouldn't this reveal some kind of inconsistency between the two theses of the manuscript? Put differently, if ds behaved as $ds(\rho, \eta=0)$, this would lead to an inconsistency with Flory theory for $ds \sim 2$, isn't it? I suggest, if my criticisms hold, that the authors perform the suggested variant of the scatter plot. If my criticisms still hold, the authors should rephrase their conclusions.

5b) Secondly, the main discrepancy between the Flory theory and the MC data is attributed to the logarithmic corrections near the UCD. Is there further evidence of these corrections to be at the origin of the discrepancy? Are there alternative explanations? Do the authors know why, for some models, it is precisely for $D=D_{ucd}$ that the long-short range analogy seems to hold (see, v.g., refs. 46,[Baños+])? In other terms: why do some models seem to exhibit such logarithmic corrections near D_{ucd} and others do not? What is the determinant of such logarithmic corrections?

6) The authors claim that $\langle \nu \rangle$.

As said before, other works have addressed the $\nu \leftrightarrow D$ connection (as it is called, for instance, in ref. 59) in models with both clean and disordered couplings, defined on both sparse and fully connected graphs. In other terms, other articles have assessed this equivalence and have found that, to some extent, for some values of ρ (and hence of $ds(\rho)$), "the spectral dimension determines the critical behaviour". This is also what the authors seem to have found: in a certain interval of ds , the critical behaviour is determined by ds .

For instance, in ref. 59, a $\nu \leftrightarrow D$ connection is found when $D(\rho) \approx 3$. In this regime, it is rather clear that, at least for such low values of ρ , and whatever the form of $D(\rho)$ and $ds(\rho)$ is in general, $ds(\rho) = D(\rho) = 2d/(\rho - d)$, and that the correction of $ds(\rho)$ and $D(\rho)$ (due to any kind of " η " away from this law is negligible. We know, hence, $ds = D$, and the critical behaviour is essentially determined by ds . As far as I understand, this is essentially what is found by the authors as well: that in some range of ds , ds plays the role of D in models defined on a regular D -dimensional lattice.

Probably the authors mean that the originality of the article resides in that the Flory and "long-range" $\nu(d)$ are compared in a whole interval of d , not only in isolated, integer values of d . I believe that the question of the originality of the present article is not very important (in any case, the authors provide the community with another representative example, that may trigger further research in the topic). Nevertheless, I would suggest the authors to state more clearly what do they mean by "correlated universality" in

\gg the dependence of the correlation length exponent ν on ds displayed in Fig.3 is the first numerical evidence of correlated universality on non-homogeneous structures

and in what precise sense

>> the study of critical indices as a function of a continuous spectral dimension appears here for the first time

or, rather, to relax the statements regarding the article's originality.

7) As far as I understand, the numerical procedure to estimate ν is algorithmic (i.e., fully automatised). Is this true? For example, the fitting window in FIG. 4 is chosen with the 5% criterion for all values of ρ : isn't it? What does "the range of N where the disagreement of the three curves remains below 5%" precisely mean? Do you mean 5% of the value ν of a given size? What is the size that is finally used to perform the linear regression in In FIG. 4? The largest size? How the errorbar of ν in figures 3,9 is estimated?

Does it represent the largest error involved in the computation (is it larger, for example than that resulting from the variation of the threshold $\text{thres}=5\%$ mentioned above)? Could the authors specify explicitly whether the procedure is completely automatic, and what is the algorithm in all its details (in the form of a simple list, or a pseudo-code, and/or a repository)?

In the same line I would suggest, if I may, to explain better the "triplets" algorithm for the estimation of d_s .

If FIG. 7 contains the same information as in FIG. 2, I would suggest to remove it.

If FIG. 9 contains the same parametric information than in FIGS. 2,3, I would suggest to remove it.

(8) I would suggest to rephrase in a more precise sense the aim of the article at the end of sec. I. The numerical analysis regards a peculiar statistical model, not stricto sensu "a continuous $O(N)$ field

theory". What is exactly the manuscript's aim and thesis? That a given statistical model will exhibit the same critical behaviour when defined in ALL graphs with common spectral dimension (something that the article does not address)? Or, rather, that

>> the universal scaling of microscopic statistical mechanics models on non-homogeneous structures is indeed described by an appropriate quantum field theory (QFT) in fractional dimension

an aim that does not refer to the concept of spectral dimension, and that the article does not address either.

In summary, I believe that the authors should describe more precisely the peculiar point that they are addressing, their motivation and the eventual controversies to be addressed, and the precise aim and reach of their contribution. I believe that most of the doubts that I raised above could be dissipated, without any change in the results, if the authors clarified in a less ambiguous way their aim and conclusions.

*

Some minor, subjective suggestions:

- Regarding the sentence <<Most numerical investigations of critical phenomena on complex networks focus on small-world networks, where critical fluctuations are Gaussian>>. Do small-world networks always exhibit gapped spectrum, or a spectral dimension $d_s = \infty$ for all values of the small-world parameter, say p_{sw} (so that the fluctuations of any model on them are Gaussian)? If so, and even if this is obvious for the authors, I guess it is not for all readers. I would suggest to provide some further explanation. In the same spirit, I suggest to clarify what is meant by "correlated scaling behaviour".

- I have found that sec. II is difficult to read, specially for non-specialists, and given the journal. For instance, the digression on the disorder correction to η , and the eventual equality between η

and the anomalous scaling exponent on the microscopic model defined on the graph (see my above criticism), may be confusing for a "mode" statistical physicist.

- <<Our conjecture is that any critical model whose microscopic components occupy the EDGES>>

I guess that the authors mean <>.

In this sentence, perhaps it would be useful to specify in what precise terms the spatial-graph analogy is meant (large-N limit, average over disorder, order of the thermal-vs-graph realisation averages, etc.).

- I would add a third limitation to the limitations i and ii to the evidence in favour of the article's main thesis (as far as I have understood it). The third, and most evident limitation is that the agreement between "both ν 's" occurs only for $d \lesssim 3$. While this may be occur for the reasons that the authors have explained, this fact actually hinders the evidence in favour of $D=d$ for all d 's. The discrepancy for $d > 3$ is striking in front of the discrepancy with the $\nu_{\text{simulation}}(D=3)$.

- I would not say that <<[...] these rigorous mathematical results on the existence of critical points on the LRDG have been confirmed numerically in Refs. 46,47.>> I would rather say that the numerical results therein are consistent with such rigorous results.

*

Even minor comments:

- I would suggest to refer to "numerical" and not to "exact" (unless the authors refer to an exact solution) in the sentence <<mismatch from the exact value in $d = 3$ >>.

- Wouldn't it be more appropriate to talk about "real" or "non-integer" dimension instead of "fractional"?

- I would suggest to modify the adjective "typical" in the first paragraph (since the referred statement has well defined conditions).

- I would suggest to substitute "universal observations" by "observations" in the second paragraph.

- Is it expected in Nature Communications, as in many other journals, to outline the article's structure at the end of the Introduction? If it is, I would recommend to do so.

- I would remove or explain the adjective "simple" in the first sentence of sec. IV. I would rephrase as well the sentence <<However, the LRDG is merely a simple graph with constant couplings and, therefore, it rather corresponds to a nearest neighbour model on a complex topology>>.

The present manuscript is about showing, numerically, universality on non-homogeneous structures. To prove their findings, authors use the self-avoiding random walk on a long range diluted graph with a futuristic thought of using LRDG to study fully connected long range systems. Authors perform Monte Carlo simulations to compute the mean squared displacement $\langle R^2(N) \rangle$ for a suitable region. Authors themselves have stated that this model is somewhat similar to a nearest neighbour model, but with some complexity involved in the topological aspects. The reason for choosing the LRDG is that the authors can tune the spatial dimension according to the demand of the numerical simulation. The manuscript is a nice piece of research, in my opinion, and it may complement studies on universality in complex structures. I also think that there are no studies have been carried out, yet, to show universality beyond regular Euclidean lattices. Moreover, the appearance of critical indices as a function of spectral dimension is also new in the case of a non-homogeneous system. I find the manuscript very useful, as it may open up new venues, and the presentation is very simple and captivating. I have no problem in accepting the manuscript for a publication in Nature Communications. However, I would like the authors to address my questions and resubmit the manuscript with appropriate changes. Then, I shall see the rebuttal and accept it.

1. I understood that this model is valid for short random-walks. If the authors would like to correct me with a general proof, they are welcome. Simple random-walk is a Markovian process, so how about the SARW used here? Are there any non-Markovian aspects involved? How do you account for the nearest neighbour coupling in this model? Is it possible to obtain a diffusive behaviour in the thermodynamic limit? How do you account for Ergodicity in this model (if possible)?
2. Is it possible to show how this model works on a honeycomb/hexagonal lattice?
3. What is the difference between using a renormalization group approach and the Monte Carlo (used here) in the context of the model presented in the manuscript?
4. Does it apply in the context of percolation theory?
5. How do you account for Quantum Phase Transition here? As a limiting case of something?
6. What is the nature of the SARW on LRDG here? Ballistic or Sub-Ballistic?
7. Authors have not discussed about the free energy and related issues. I would like to see a short discussion about free energy in the present context.

We sincerely appreciate the reviewer’s comprehensive and insightful review of our manuscript entitled “Universal scaling in fractional dimension”. The detailed comments and inquiries are invaluable, and we are committed to addressing each point the reviewer has raised. In the following we report a point-by-point response to the reviewer’s queries and concerns:

- 1 Reviewer: “One of my main doubts regards the discussion on the spectral dimension in equation 6...”

We acknowledge that the previous version of the manuscript was not sufficiently clear concerning the issue of the relation between our findings and the traditional picture delineated for long-range critical systems. As the reviewer correctly points out, the interpretation of η as a model-independent property of the graph may appear contradictory with the existing literature. Most of the current literature, including the paper [Baños et al, Phys. Rev. B 86, 134416 (2012)] mentioned by the reviewer (Ref.[20] in the revised manuscript), is concerned with the comparison between fully connected long-range systems, i.e., those where the coupling matrix scales as $r^{-\alpha}$, and local ones. In this case, it was noticed that the critical exponents of the long-range model roughly coincide with the ones of the local system in an effective dimension

$$d_{\text{eff}} = (2 - \eta_{\text{sr}})d/\sigma, \quad (1)$$

where η_{sr} , as noted by the reviewer, is the anomalous dimension of the statistical model under study. This is very different from our Eq. (6), where the spectral dimension may (or may not) receive a disorder dependent correction, which is a property of the graph alone. So, the reviewer is perfectly correct in their summary of our results.

However, the reviewer will note that the effective dimension result of Ref. [20] is only an approximate one and has been widely shown not to be exact, see for example one of the authors’ previous work Ref. [22], but also exact theoretical studies made in the new Ref. [23]. This approximate nature is demonstrated by extensive studies for the long-range Ising model, see Fig. R1, where the correlation length exponent of a fully connected long-range Ising model is compared with its nearest neighbour counterpart at the same effective dimension. All data points in Fig. R1 have been obtained with conformal bootstrap and have negligible uncertainties on our scale.

The above arguments (together with Fig. R1) demonstrate that the effective dimension approach discussed in several references including [Baños et al, Phys. Rev. B 86, 134416 (2012)] is a useful, but approximate relation.

The reason behind the approximate nature of d_{eff} is easily understood by field theory arguments. The propagator $G(q)$ of a long-range field theory in the continuum shall obey a relation of the form

$$G(q)^{-1} \approx p^\sigma + Z p^2 + W p^{2-2\sigma} + \dots \quad (2)$$

Figure R1: Correlation length exponent of the Ising model. Black squares are for a regular lattice with nearest neighbour couplings. Red circles are for a regular lattice with fully connected long-range interactions. Both results have been obtained via conformal bootstrap, see Refs. [El-Showk, Phys. Rev. Lett. 112, 141601 (2014); Behan et al., arXiv:2311.02742].

while the propagator of a local theory contains only analytic terms

$$G(q)^{-1} \approx p^2 + W p^4 + \dots \quad (3)$$

So, a complete analogy between the two theories is impossible as the operator content of the long-range theory is far wider than that of its local counterpart.

Our model is different, as it is not an actual long-range theory, but rather a local theory whose components are embedded in a complex geometry whose spectral scaling is reminiscent of a long-range spectrum. Indeed, in our model all of the couplings (hopping elements) have the same strength and the spectral properties are modified only through the distribution of the couplings over the graph. Thus, the scaling behaviour of a critical system on the LRDG graph is expected to be fully determined by the spectral dimension of the graph alone, without the interplay between spectral and critical scaling which for clean, fully connected long-range interactions is represented by the η_{sr} correction in the expression $d_{\text{eff}} = (2 - \eta_{\text{sr}})d/\sigma$.

As a further evidence of the inconsistency of Eq. (1) with our studies, the reviewer may note that the correlation length exponent value $\nu_* \approx 3/4$, which is found in local systems, is recovered at $\sigma_* \approx 2$ by our numerics. This is consistent with our estimate $d_s(\rho = 4) \approx 2$. On the contrary, the effective dimension relation in Eq. (1) would yield $\sigma_* = 2 - \eta_{\text{saw}}(d = 2) = 1.792$, see Fig. R2.

In the revised version of the manuscript the difference between our expression for d_s and the

Figure R2: SAW correlation length exponent on the LRDG as a function of $\rho = 2 + \sigma$. The nearest neighbour value is recovered at $\sigma_* \gtrsim 2$ consistently with the spectral dimension estimate $d_s = 2d/\sigma$, but not with Eq. (1), which would predict $\sigma_* = 1.792$. The dashed curve is the Flory theory in dimension $d = 2d/\sigma$.

effective dimension approach developed for fully connected long-range interacting systems is properly discussed.

2 Reviewer: “The authors claim that the spectral dimension d_s of the 2D LRDG is as in eq. (6) with $\eta = 0...$ ”

We apologize to the reviewer if some of our comments regarding the correction η may have sounded confusing. Our statement was made in connection with our previous publication, Ref. [50], where we observed that the purely geometric correction η in $d = 1$ as a function of d_s follows roughly the same behaviour of the anomalous dimension of the scalar ϕ^4 theory as a function of the space dimension d . Thus, we conjecture that the η correction in $d = 1$ originates from a similar mechanism as in typical statistical field theories, which do not allow for $\eta < 0$. In any case, we concur with the reviewer that, given the lack of substantial evidence in this direction, this comment is confusing rather than clarifying. In the revised version of the manuscript we amended the comment and replaced it with a discussion of the numerical evidences in favour of $\eta = 0$. Our discussion points out that a negative shift $\eta < 0$ is likely caused by the nonlinearity observed in the finite-size scaling at $\rho \simeq 4$.

3 Reviewer: “The authors claim, furthermore, that $\eta = 0$ in eq. 6 is consistent with reference 46. However...”

The reviewer is clearly right about the two models being different and we thank the reviewer for pointing out that our comment was confusing. However, both our model and the mentioned reference share the same spectral dimension. In fact, the spectral dimension is a universal quantity and should not depend on the microscopic construction of the model, see Ref. [14]. In Ref. [50] we have shown that the spectral dimension for the one-dimensional version of our model is independent of the average connectivity and that models with a different prefactor for the probability distribution share the same spectral dimension. In other words, if one modifies the activation probability for the graph edges to

$$P = p/r^\rho \quad (4)$$

with $p \neq 1$ (note that in our paper we have $p = 1$), we expect to have the same spectral dimension for any p large enough to obtain a macroscopic cluster.

As the reviewer correctly noted, the authors of Ref. 46 (now [52]) take a different approach and fix the total number of links N_{tot} . From the perspective of statistical mechanics p and N_{tot} are conjugate variables and having a fixed N_{tot} corresponds to having a p which fluctuates during the extraction processes. Hence, the two models are related to each other by a change in their ensemble description. Due to the universal properties of the spectral dimension discussed in Ref. [32] as well as the numerical evidence obtained in Ref. [50], we expect the two models to share the same spectral dimension.

Our comment has been slightly modified to evidence this fact. However, given also the further comments of the reviewer we have decided to revise our statements concerning the exact value of the correction η as well as the relation to Ref. [14].

4 Reviewer: “As the authors perfectly know...”

Yes, there are several references studying the correspondence between the universal scaling of long-range and nearest neighbours systems. Most of these references consider the case of fully connected long-range interactions, where the equivalence is clearly approximate as demonstrated by Fig. R1, see also the reply to comment 1 of the reviewer.

In the disordered case, studies have been less systematic and have mostly considered the case of sign-changing interactions, this is the case of Ref. [20] mentioned by the reviewer, but also older studies such as Ref. [56] [Kotliar, et al. Phys. Rev. B **27** 602 (1983)]. Sign-changing interactions induce a wide range of different phenomena, such as frustration and glassy behaviour. Thus, as for the case of fully connected interactions, the weighted couplings generate a non-trivial interplay between the spectral fluctuations of the geometric structure and the statistical fluctuations of the model.

In this perspective our model is unique. All elements of our coupling matrix have the same value and, thus, encode a purely geometric information. Thanks to this simplicity, we expect the LRDG to be fully described by the spectral dimension and to constitute the first realization of a local model in noninteger dimension. Our results in this direction are promising; indeed, the SAW scaling does not exhibit the traditional interplay between the spectral scaling and the model-dependent anomalous dimension. In fact, we find $\sigma_* \approx 2$ in our model instead of $\sigma_* \approx 2 - \eta$ for fully connected long-range interactions, see Fig. R2. Also, our results are unique, since to our knowledge, no previous analysis of the critical exponents exists for a model where geometry and statics are as transparently separable as in our case.

5a Reviewer: “Perhaps my main doubt regards the results in FIG. 3...”

We understand why the reviewer may be puzzled by the agreement of our numerical curve with the Flory prediction, although we have evidenced a seemingly systematic correction appearing for d_s at $\sigma \approx 2$. However, we believe this agreement has a simple explanation. Our manuscript provides substantial evidence towards two general statements:

1. The universal properties of a critical system on a diluted graph in the absence of frustration depend only on the spectral dimension.
2. The universal scaling as a function of d_s coincides with the one of a continuous theory with $d = d_s$.

Given statement (1) it seems natural to expect a cancellation between the systematic finite-size correction in the graph structure and the one in the critical scaling, once the results are reported as a function of the spectral dimension. Indeed, the numerical simulations of the critical model under study occur on a finite graph and probe the “finite-N” spectral dimension. Thus, while the estimate for d_s of the finite graph is shifted upwards with respect to the theoretical prediction for the infinite graph as a function of σ , the numerical estimate for ν is shifted downwards with respect to the analytic Flory theory in $d_s = 2d/\sigma$ (coherently with the result in a larger spectral dimension), see the result in Fig. R2 (note that in Fig. R2 the result for $1/\nu$ is shown, so that a downward shift of ν appears as an upward shift with respect to the Flory estimate). Therefore, the net effect as a function of d_s is very well compensated.

Figure R3: SAW Correlation length exponent on the LRDG as a function of d_s .

In any case, the overall effect is quantitative rather than qualitative as the reviewer may verify in Fig. R3, where we report the numerical estimate of ν as a function of both the numerically estimated d_s and the (conjectured) theoretical prediction $d_s^{\text{th}} = 2d/\sigma$. The region $d_s \approx 3$ is untouched, while in the region $d_s \approx 2$ the effect is mild since the originally reported data were slightly underestimated with respect to Flory theory, while they become overestimated if reported as a function of the theoretical spectral dimension.

We included a comment in the manuscript discussing the cancellation occurring between the systematic finite-size error found for d_s and ν . We retain our presentation of the data for two reasons: i) generating the curve solely by the numerical estimate appears more consistent with our statement (1) above; ii) using the numerical estimate of d_s avoids any unnecessary bias towards the $\eta = 0$ conjecture, whose role in the present version of the manuscript is now less emphasized, in agreement with the previous request of the reviewer.

5b Reviewer: “Secondly, the main discrepancy between the Flory theory and the MC data is attributed to the logarithmic corrections near the UCD. Is there further evidence...”

Figure R4: The magnetic critical exponent β of the quantum long-range Ising model in $d = 2$, taken from Ref. [Kozioł et al. Phys. Rev. B 103, 245135 (2021)]. The appearance of logarithmic corrections at the upper critical dimension generates substantially larger uncertainties close to $\sigma \approx 1$. The model is fully connected and there is no ambiguity on the applicability of the relation $d_{\text{eff}} = 2d/\sigma$ at $\sigma \approx 1$. Yet, the numerical simulation suffer large errorbars in the vicinity of the upper critical dimension.

On this point, we feel the reviewer misunderstood the meaning of our statement on logarithmic corrections. It is true that for long-range (possibly diluted) models in the mean-field limit, i.e. $d_s \rightarrow d_{\text{ucd}}$, one retrieves the exact equation $d_{\text{eff}} = d_s = 2d/\sigma$, since both the graph and the model anomalous dimensions, i.e. η and η_{sr} vanish. However, to numerically verify such limit is notoriously hard since the fitting function for critical quantities close to the upper critical dimension contain logarithms of the system sizes, due to the fact that the RG flow between the Gaussian and the interacting fixed points in the coupling space becomes marginal, a modern viewpoint on this problem is found in Ref. [SciPost Phys. Lect. Notes 60 (2022)]. The logarithmic corrections do not alter the universal picture in the thermodynamic limit, which, again, is consistent with $d_{\text{eff}} = d_s = 2d/\sigma$, but make it hard to verify numerically.

The difficulty of numerically estimating critical exponents above the upper critical dimension is consistent with several other numerical studies, especially on the Ising model, see Fig. R4. In the revised version of the manuscript we included a sentence citing the Ref. [SciPost Phys. Lect. Notes 60 (2022)] and clarifying why we believe the only issue is the presence of logarithmic corrections.

6 Reviewer: “The authors claim that \ll the study of critical indices as a function of a continuous spectral dimension appears here for the first time \gg ...”

We understand that the reviewer may have found our sentences too grand, we just wanted to emphasize the uniqueness of our model. Indeed, as we discussed before, differently from all previous references, we study a model where the tuneable spectral dimension arises purely due to the geometry, without competing effect such as frustration or glassy behaviour and, possibly, without the conventional interplay between the (model specific) statistical fluctuations and the disorder fluctuations. We have modified each sentence mentioned by the reviewer as follows:

- 1-old “the study of critical indices as a function of a continuous spectral dimension appears here for the first time”
- 1-new “While here, for the first time, the entire curve of a critical index is reported in the spectral dimension range $d_s \in [2, 4]$, where the universality is expected to be correlated, i.e., non mean-field”
- 2-old “the dependence of the correlation length exponent ν on d_s displayed in Fig.3 is the first numerical evidence of correlated universality on non- homogeneous structures”
- 2-new The above sentence has been deleted.

7 Reviewer: “As far as I understand, the numerical procedure to estimate ν is algorithmic (i.e., fully automatised). Is this true? For example...”

The reviewer is correct in their understanding of the algorithm and the fact that the fitting process is fully automated, the only difference is for values of $d_s > 3$ and $d_s \geq 3$. In the following we report a simple list with the passages that have been performed on the data:

1. Let us introduce the nomenclature

$$f_L(N) = \frac{d \log \langle R(N)^2 \rangle}{d \log N}$$

where N is the length of the walk and L the linear size of the LRDG under consideration. Then, the fitting window is defined as the region of N where $|f_{L_1}(N) - f_{L_2}(N)| / |f_{L_1}(N) + f_{L_2}(N)| \leq 0.05$ for all considered values of L_1 and L_2 .

2. Once the fitting window is determined the fit is performed on each curve extrapolating a different, but consistent value of ν for each of the curves.

3. In the region $\rho \geq 3.2$ (where the $f_L(N)$ curves show a substantially linear behaviour) we report the average value of ν between all the curves, while in the region $\rho < 3.2$ we consider only the value of ν obtained from the largest system size.
4. The errorbar is obtained as the maximum difference between the extrapolated values of ν for all values of ρ .

Variations of the fitting window within 3-7% do not substantially alter the estimates nor the errorbars. A similar list has been included for the derivation of the spectral dimension. Fig. 7 and Fig. 9 have been removed as suggested by the reviewer.

8 Reviewer: “I would suggest to rephrase in a more precise sense...”

We thank the reviewer for pointing out that the scope of our paper needs clarification. We hope that after our extensive reply to all the reviewer questions our idea may appear more clear.

As the reviewer knows, all previous efforts to devise an exact correspondence between the universal behaviour of models with the same spectral scaling, i.e., the same spectral dimension, was not successful. We attribute this failure by the previous studies, which consider models in presence of glassiness and frustration, or in the case of fully connected long-range interactions, to the difficulty of properly separating the contributions from the geometry from those of the interaction.

From this perspective comes the uniqueness of our model, whose information is purely geometrical (all couplings have the same sign) and it may really be considered as a nearest neighbour model on a complex geometry. Thus, according to our statement (1) the model’s universality is fully determined by the spectral dimension and, thus, its thermodynamic limit shall correspond to a quantum field theory where the dimension of the continuous space is set to $d = d_s$. This is the spirit of the Flory prediction; it does not make reference to any peculiar discrete lattice/graph, it is only derived via scaling arguments as it is usually done for continuous field theories. Yet, it provides very good agreement with our numerical data, showing that the behaviour of the critical indices on the LRDG is indeed universal and can be obtained by a field theory in continuous space whose dimension corresponds to the spectral dimension of the underlying graph.

Since the reviewer found our formulation of this line of thought unclear, we have substantially revised our introduction.

9 Reviewer: “Some minor, subjective suggestions: ...”

- Despite some evidence that small-world networks always exhibit Gaussian critical fluctuations, we do not have an exact argument. Therefore, we have mildly weakened our sentence.
- The discussion mentioned by the reviewer has been reduced and the entire section has been streamlined.
- The word edges has been corrected.

- While we understand why the reviewer may consider the disagreement at $d_s > 3$ problematic, we believe that the difficulty to identify a precise slope for the fit function at $\rho \leq 3.1$ gives a strong enough indication that the disagreement is caused by numerical issues rather than a fundamental one and we will rather not include it in that list.
- the sentence has been modified as requested by the reviewer.

10 Reviewer: “Even minor comments: ...”

All minor comments have been properly addressed.

Reply to Reviewer 2 – 1692661074/Bighin

We thank the reviewer for the time spent in reading our manuscript and for their positive evaluation of our work. In the following we reply to their comments one by one and describe the modifications performed to the manuscript in order to clarify those points.

- 1 Reviewer: “I understood that this model is valid for short random-walks. If the authors would like to correct me with a general proof, they are welcome. Simple random-walk is a Markovian process, so how about the SARW used here? Are there any non-Markovianity involved? How do you account for the nearest neighbour coupling in this model? Is it possible to obtain a diffusive behaviour in the thermodynamic limit? How do you account for Ergodicity in this model (if possible)?”

We thank the reviewer for the insightful question. First of all, we would like to clarify that our definition of self-avoiding random walk (SAW) follows Ref. [35]. Thus, our model is not the true self-avoiding walk and is certainly not a Markovian process. Moreover, our model is not strictly speaking a stochastic process either, see the introduction of Ref. [35].

The interaction between neighbours follows the prescription described above Eq. (7). At each step the walk is enlarged/reduced by one unit choosing with equal probability within all possible walks of the desired length. If the selected move generates a self-intersecting walk or a 0-length walk, then it is rejected and the old configuration is considered again. Below Eq. 7 we also clarify that this overall process is ergodic and that the sampling probability for a walk of length ω is given by Eq. 8.

In our definition of the SARW the model becomes diffusive only for $d_s \geq 4$, where the excluded volume is negligible. For all other values $2 \leq d_s < 4$ the behaviour is super-diffusive since $\nu > 1/2$ in Eq. (3). In the beginning of Sec. II we introduced a small paragraph describing our model in more detail and clarified the issue of Markovianity and diffusivity.

- 2 Reviewer: “Is it possible to show how this model works on a honeycomb/hexagonal lattice?”

For the strictly nearest neighbour model the universality class on the hexagonal lattice will correspond to the one of the square case, See Ref. [J. Stat. Phys, 45, 459–470, 1986]. On the other hand, a full generalization of the dilution process to an hexagonal lattice embedding

will need a simulation campaign lasting several months: one would need to first depict the spectral dimension curve as a function of σ and then evaluate the exponent ν for different values of σ .

In principle, we do not expect anything to change as the spectral dimension is expected to be a universal quantity and, as such, it shall not depend on the lattice embedding. On the other hand, the hexagonal lattice may modify the subleading corrections and possibly improve the scalability of the model. We leave this interesting question to future work, but we included a comment in the manuscript describing this issue.

3 Reviewer: ‘What is the difference between using a renormalization group approach and the Monte Carlo (used here) in the context of the model presented in the manuscript?’

We thank the referee for this very interesting comment. The choice of the SARW was partially due to the possibility to access a simple and reliable theoretical estimate such as Eq. (5).

Figure R5: The correlation length exponent ν of $O(\mathcal{N})$ field theories in the $\mathcal{N} \rightarrow 0$ as a function of the continuous space dimension $d \equiv d_s$.

Another way to validate the correspondence between the discrete model and the continuous theory would be to use renormalization group arguments. Unfortunately, given the approximate nature of the RG approach, it is very difficult to obtain reliable estimates of the critical

exponent ν in $d < 3$. Indeed, it has proven hard to construct a set of RG equations capable of accounting for the conformal nature of the 2D problem and, so, reproduce the exact estimate given by Flory theory in 2D.

Indeed, we had performed an RG study of the problem using functional RG, see the red empty points in Fig. R5, but the obtained value of the correlation length exponent ν does not lie on top of the numerical or theoretical Flory curve for $d_s \lesssim 3$. This is solely due to the approximate nature of the RG equation which, in fact, does not reproduce the conformal estimate in $d = 2$ (empty squares in Fig. R5); a similar effect is observed also in the Ising model, see Ref. [55].

A comment on this aspect was introduced in the conclusion of the revised version of the manuscript.

4 Reviewer: “Does it apply in the context of percolation theory?”

The case of percolation theory is much more involved than the problem of the SARW. Indeed, percolation does not allow for a simple field theoretical description, thus making it hard to have theoretical estimates for the continuous field theory in $d \in \mathbb{R}$. In a previous paper, we have addressed the correspondence between the percolation problem in different effective dimensions, see Ref. [81]. However, these studies were performed in $d = 1$ and could only make use of perturbative expressions for the universal critical exponents. In this perspective the study presented here is much more complete.

5 Reviewer: “How do you account for Quantum Phase Transition here? As a limiting case of something?”

We do apologize to the referee if our discussion has led to some confusion. In our paper we only deal with classical phase transitions. However, we do employ the quantum field theory description for the continuous theory, see Eq. (1). Yet, our discussion only makes reference to quantum field theory in Euclidean space that, due to the quantum to classical correspondence, is used to describe a classical phase transition.

We included a comment to clarify this issue in the revised version of the manuscript.

6 Reviewer: “What is the nature of the SARW on LRDG here? Ballistic or Sub-Ballistic?”

It can be exactly proven that the SARW scaling is ballistic in $d = 1$. In the reply to the first comment of the reviewer, we have clarified that the SARW scales diffusively for $d_s \geq 4$. Thus, as a function of d the SARW interpolates between perfect ballistic behaviour in $d = 1$ and diffusive behaviour in $d \geq 2$. We have added a comment after Eq. (5) which clarifies this aspect.

7 Reviewer: “Authors have not discussed about the free energy and related issues. I would like to see a short discussion about free energy in the present context.”

The free energy is not a universal concept. Therefore, it evades the correspondence discussed in our paper. For example, it is well known that the Flory theory can be predicted via free-energy arguments, but the value of the free energy itself is greatly overestimated within the Flory approach, see Ref. [A Renner. Self-avoiding walk models for non-random heteropolymers. Thesis, 1996]. In any case, some claims of universality have been made for the free energy of $O(N)$ models at the critical point and it would be of great interest to verify this conjecture on a non-homogeneous graph. We have added a comment on this interesting possible studies in the conclusions.

REVIEWER COMMENTS

Reviewer #1 (Remarks to the Author):

This is a second report on the manuscript "Universal scaling in fractional dimension" that has been submitted for publication in Nature Communications.

After having read the authors' answer to my first report, I have understood better their aim and theses. First, I realise that the manuscript is now clearer and less ambiguous. Second, the rebuttal letter has reaffirmed my original opinion:

1. I believe that the manuscript brings interesting and representative results, that could trigger a revival interest in the topic and, consequently, that they would deserve to be published in a high-impact journal as Nature Communications...

2. if only presented differently, since, in my humble opinion, the article's results are overinterpreted. High-quality research standards impose the conclusions to be drawn beyond a reasonable doubt from the results, or to be presented as mere conjectures, conditioned on clearly stated hypotheses, still to be verified.

In the following, I will argue in what precise sense I believe that the results should be presented differently.

Again, my criticisms do not regard the results/calculations, but their interpretation and presentation only. Furthermore, I believe that ****minor modifications**** should be enough --albeit necessary. I am rather sure that the authors will agree on the spirit of my below criticisms and, consequently, that a further correction from me will not be necessary.

I sincerely hope to have been useful for the authors and for the readers.

*

In the following, I will outline the manuscript results as I understand them; the conclusions that, to the best of my understanding, **could be** drawn from them; the hypotheses that would be needed to draw such conclusions from the results.

The outline of results:

R1. The authors identify a statistical model (the SARW) for which there is a very accurate theoretical relation, derived from scaling arguments, $\nu_F(d)$.

R2. They identify as well a self-averaging ensemble of networks (LRDG) for which the spectral dimension can be numerically estimated.

R3. This setup is peculiar and interesting in what: the network of interactions is irregular (sampled from an ensemble) but there is no quenched disorder in the interaction strength (so that one can isolate both sources of stochasticity); one can compare $\nu_{\text{num}}(ds)$ vs $\nu_F(d)$ in a wide range of values of ds and d respectively.

R4. Comparing the theoretical relation $\nu_F(d)$ vs d , with the numerical one $\nu_{\text{num}}(\sigma)$ vs $ds(\sigma)$, **a relative agreement is found**. In particular, it is found good agreement for ds in a neighbourhood of $ds \sim 3$ [when the agreement is less surprising, since $ds \sim D$ for $ds \sim 3$].

Up to here, there is, roughly speaking, what can be concluded beyond a reasonable doubt (up to the influence of the numerical protocol). Beyond this, one enters into the world of the speculative interpretations.

One **could try** (it is NOT the only interpretation!) to interpret these results drawing the following conjectures:

The agreement $\nu_{\text{num}}(ds)$ vs $\nu_F(d)$ is relative: for high ds , there is disagreement. For low ds , there is agreement, albeit one knows that the estimation of ds suffers from significant finite-size effects. However, one may conjecture that,

- (c1) first, the discrepancy for high values of ds is due to finite-size effects and it cancels for large N ;

- (c2) second, the finite-size effects of d_s (for low d_s) compensate with further finite-size effects of $\nu_{\text{num}}(\sigma)$, in such a way that both cancel for low N , and keep cancelling for large N .

Under the conjectured hypotheses c_1, c_2 , it would result that $\nu_{\text{num}}(d_s)$ behaved as $\nu_F(d)$, despite there is no clear numerical evidence on this. We would have, hence, that the critical behaviour of **a particular statistical model** (the SARW), or at least one of its critical exponents, would be governed by the spectral dimension only. Now, further supposing that

- (c3) not only the SARW but any other statistical model (satisfying certain properties), if defined on a LRDG, actually exhibited the same conjectured behaviour: its critical exponents depended on d_s as predicted by equation (1) taking $d=d_s$,

one would have (jointly supposing c_1, c_2, c_3), that the critical dimension of certain models **in the LRDG** is governed by the spectral dimension. Now supposing that

- (c4) the same happened for all the statistical models in all ensembles of (self-averaging?) graphs,

then one could state, roughly speaking, what the authors call statement 1 in p. 1:

S1) the critical dimension of certain models is governed by the spectral dimension.

This statement would apparently contradict other studies, that have found that it is actually necessary to correct the "effective dimension" D with a **model-dependent quantity** $\eta_{\text{sr}}(D)$. Or, in other words, that the spectral dimension is not enough to govern the critical behaviour of systems defined in non-regular graphs. However, further conjecturing

- (c5) that such apparent disagreement was attributable to the all-to-all character of the considered interaction networks, and to their quenched disorder...

this would finally resolve the apparent contradiction, and would permit to conjecture S1.

A rather critical reader could think that the conjecture S1 requires a number of underlying assumptions (at least 5), that is larger than the number hints brought to evidence in favor of S1. However, such balance may be subjective. I would perfectly admit the possibility of S1-2 being true (indeed, the dependence of the critical behaviour on \mathcal{N} being all through eq. (1), rather than through an ad hoc correction to $D(\sigma, \eta_{sr})$, is a general, hence suggestive possibility).

I consider S1 a licit conjecture ****if only stated as such****. If the observations/results R1-4 were presented as not being more than they are, and the conjecture S1 were presented as such, and requiring the (explicitly stated) underlying above conjectures c1-5 to be true, the article would be already excellent and interesting enough, and would gain much in falsifiability! The fact that the results are somehow ambiguous and that more than one interpretation is consequently possible, makes the article even more interesting. The interpretations, however, must be clearly separated from the results, or the article will risk not complying with research best praxes. I disagree, in other words, with the sentence:

>> The present work provides substantial evidence for two general statements [...]

or with

>> [...] and show that it depends only on the spectral dimension ds .

Since it is clear that the present work considers one single statistical model in a particular kind of graph, and that it is necessary to interpret the deviations of ν_{num} with respect to its expected behaviour ****in a precise sense****, in order to conclude that ν is governed by ds ... only in the model and in the graph family at hand! I would rather say that the article brings an interesting case of study that is, in principle, CONSISTENT WITH S1-2, under certain assumptions. And I would change the article text accordingly. By stating more prudently the conclusions, it would become much more robust. The verb "to show" clearly has a different meaning, specially in theoretical physics.

By the way, some reader may miss a more explicit formulation of the statements 1-2 in p. 1.

- What are the characteristics that the model has to satisfy? Being describable by a $O(\mathcal{N})$ field theory with integer d when N is large and it is defined on a d -dim regular lattice?

- Afterwards, what are precisely the characteristics of the graphs for which S1-2 apply (self-averaging --in what precise sense?--, connected, dense/sparse, well-defined ds , non-weighted...)?

- Afterwards, the claim S2 states that $\nu_{\text{num}}(d) = \nu(d)$ where $\nu(d)$ is the one obtained from eq. (1) but... in what precise sense? In the sense of ref. 8, I guess. In the article, $\nu(d)$ is taken as in equation (5), derived "from scaling arguments" but, in a general setting, and in practical terms, how one would compute $\nu(d)$ from eq. (1) for non-integer d ? For example, how could one estimate $\nu(d)$ for Ising or Heisenberg classical models, and using what kind of approximations? How accurate are such approximations?

As further, minor comments,

- I find that the first paragraph of sec I keeps not being clear. I believe that, for most physicists, it is not clear the connection between Gaussian fluctuations or mean field behaviour (of what?) and small-world networks.

- In my opinion, discussing the discrepancy of the numerical Flory exponent in $d=3$ (of order ~ 0.02 in $1/\nu$), in front of a discrepancy 5 or 6 times larger, of ν_{num} with respect to ν_F , is pointless. I would do the same criticism regarding the list (i), (ii) in sec. IV. The fact that there exist finite size effects in the determination of ν_{num} **does not imply "straightforwardly" that, in absence of such finite-size effects, one would get a certain conjectured behaviour. Even if there were no reason to believe the opposite, nor alternative conjectures, even if it turned to be true with more accurate methods, this would not modify my criticism. This is the reason for which one actually performs experiments and calculations, isn't it? In my opinion, the beauty and power of physics is that, once the hypotheses have been set, it is not necessary to stand on conjectures.

Reviewer #2 (Remarks to the Author):

The revised manuscript is well written and clean. Authors have addressed all my concerns in the previous report. The revised version is more explanatory and inclusive. I accept the revised manuscript in its current form for a publication in the Nature Communications Journal. I personally congratulate the authors for their hard work.

Dear Referee,

Thank you for the insightful feedback you provided in your second report on the manuscript titled “Universal Scaling in Fractional Dimension,” submitted for publication in Nature Communications.

Your constructive comments and critiques have been instrumental in refining the clarity and presentation of our work. We appreciate your positive evaluation of the manuscript’s potential to spark renewed interest in the subject matter.

Moreover, we acknowledge your concerns about the interpretative aspects of the results and your suggestion for presenting them differently. Your detailed breakdown of the results, conclusions, and underlying hypotheses was immensely helpful. We understand your emphasis on avoiding over-interpretation and the importance of clearly distinguishing between results and speculative interpretations.

Your proposed modifications, particularly the suggestion to state conjectures explicitly and link them to specific hypotheses, were closely followed. In the new version of the manuscript, each step of our interpretation is accompanied with a clear statement of the underlying assumptions, which have been labeled according to the referee notation. In particular, conjectures C1 and C2 are now explicitly stated in the last part of Sec. III (specifically the first column of page 6). The remaining three hypotheses C3-5 are spelled out in the conclusions, where we also discuss how and under which limits our results support statements S1 and S2. Your additional comments on the characteristics of the model and the need of further clarifications of statements 1-2 have been also implemented, see the text highlighted in blue below the list with items S1 and S2. In particular, we have adressed the following questions:

- *Referee: “What are the characteristics that the model has to satisfy? Being describable by a $O(N)$ field theory with integer d when N is large and it is defined on a d -dim regular lattice?”*

Yes, in order to behave according to S1-2 the microscopic model under study has to obey a Landau theory description in the form of $O(N)$ models close to the critical point. A comment on this point has been made after statement S2.

- *Referee: “Afterwards, what are precisely the characteristics of the graphs for which S1-2 apply (self-averaging –in what precise sense?–, connected, dense/sparse, well-defined ds , non-weighted...)?”*

The graphs described by statements S1-2 shall possess a well defined spectral dimension, which describes both the local and average behaviour, see Ref. [13]. Also, the graphs shall be self-averaging, i.e. the spectrum becomes independent by the peculiar disorder realization in the thermodynamic limit. Finally, the interactions have to be unfrustrated. A comment on all these properties is included after statement S2.

- *Referee: “Afterwards, the claim S2 states that $\nu_{num}(ds) = \nu(d)$ where $\nu(d)$ is the one obtained from eq. (1) but... in what precise sense? In the sense of ref. 8, I*

guess. In the article, $\nu(d)$ is taken as in equation (5), derived "from scaling arguments" but, in a general setting, and in practical terms, how one would compute $\nu(d)$ from eq. (1) for non-integer d ? For example, how could one estimate $\nu(d)$ for Ising or Heisenberg classical models, and using what kind of approximations? How accurate are such approximations?"

The quantity $\nu(d)$ simply refers to the universal scaling index to the continuous field theory in Eq. (1) and it can be calculated in various way. In principle, one could also perform a numerical simulation of the appropriate field theory action assuming a power-law density of state $\mathcal{D}(\varepsilon)$. Then, the comparison between the outcome of the numerical simulation for the microscopic model on the graph and that of the continuous theory with power-law density of states will be enough to prove universality. Of course, several other means to estimate $\nu(d)$ exist such as conformal bootstrap, which is believed to produce exact results, but also functional RG, whose estimates of the critical indices in continuous dimension hold notable accuracy already at low level of approximation, see Ref.[7]. A comment on our notion of universality has been introduced in the conclusions of the paper.

Finally, the minor comments by the referee were also addressed:

- *Referee: "I find that the first paragraph of sec I keeps not being clear. I believe that, for most physicists, it is not clear the connection between Gaussian fluctuations or mean field behaviour (of what?) and small-world networks."*

This statement closely follows Ref.[86], see the end of Sec. IX which states: "These models have also been studied on complex networks with different clustering coefficients, degree correlations, etc. It seems that these characteristics are not relevant, or at least not essentially relevant, to critical behavior [...]. We suggest that the critical fluctuations are Gaussian in all networks with the small-world effect, as is natural for infinite-dimensional objects."

Since the referee found our summary of the previous section unclear, we have extended it and added two footnotes. The first footnote refer to Ref. [85] and clearly indicates what is our definition of mean-field and how to apply it to compute universal indices. The second footnote points the reader to Ref. [86] for a more detailed justification of the overall statement.

- *Referee: "- In my opinion, discussing the discrepancy of the numerical Flory exponent in $d=3$ (of order 0.02 in $1/\nu$), in front of a discrepancy 5 or 6 times larger, of ν_{num} with respect to ν_F , is pointless. I would do the same criticism regarding the list (i), (ii) in sec. IV. The fact that there exist finite size effects in the determination of ν_{num} **does not imply "straightforwardly" that, in absence of such finite-size effects, one would get a certain conjectured behaviour. Even if there were no reason to believe the opposite, nor alternative conjectures, even if it turned to be true with more accurate methods, this would not modify my criticism. This is the reason for which one actually performs experiments and calculations, isn't it? In my opinion, the beauty and power*

of physics is that, once the hypotheses have been set, it is not necessary to stand on conjectures.”

We agree with the referee that our discussion of the discrepancy at $d_s = 3$ may have been overstretched; for this reason we have removed the list (i), (ii) in Sec. IV and replaced it with a point-by-point discussion of conjectures C3-5 and how their justification is necessary before statements S1 and S2 can be validated.

We are confident that the referee will find the revised version of the paper suitable for publication and we thank them once again for their time, thoroughness, and commitment to advancing the quality of our manuscript.

Best regards,
Nicolò Defenu
(On behalf of all authors).